# Applying Statistical Analysis and Economics Models to Unscramble the Depositional Signals from Chemical Proxies in Black Shales

Karin Goldberg [1,*]  and Lucas Goldberg Da Rosa [2]

1   Department of Geology, Kansas State University, 108 Thompson Hall, Manhattan, KS 66506, USA
2   Department of Economics, Kansas State University, 244 Waters Hall, Manhattan, KS 66506, USA; lucasgoldberg@ksu.edu
*   Correspondence: kgoldberg@ksu.edu; Tel.: +1-785-532-6724

**Abstract:** The complex controls on the accumulation of organic-rich rocks remain elusive, despite their economic importance as source rocks and unconventional reservoirs, partially due to the multitude of factors that may impact production and preservation of organic matter in sediments. The complexity of Earth systems is comparable to the intricacies of Economics, and application of statistical and econometrics methods and models to analyze geological data may assist interpretation of the processes controlling organic burial. Chemical indices calculated for mudrock datasets from modern sediments and the Woodford Formation were used as proxies for detrital input, primary productivity, redox conditions, and upwelling, and a series of statistical analyses were run to test whether these methods were useful to discriminate different depositional conditions and establish the controls on total organic carbon (TOC) in the sediments. Model results showed that chemical proxies reliably predict not only TOC but also indicate correlations between indices. Our results suggest that detrital input, primary productivity and bottom-water anoxia are relevant drivers of organic content in the sediments, but the first two appear to have a more significant role in organic burial, illustrating the usefulness of these methods to assess depositional parameters in organic-rich rocks.

**Keywords:** organic rocks; chemical proxies; depositional environment; source rock; unconventional reservoir; data analysis

## 1. Introduction

Black shales (mudrocks that contain more than 1% total organic carbon [1]) are an essential element in petroleum systems. In conventional systems, they are sources of hydrocarbons and may act as seals, and in unconventional petroleum systems, they are also reservoirs. In the U.S., the latter accounted for 65% of the crude oil and 78% of the natural gas production in 2020, according to the U.S. Energy Information Administration.

In addition to the importance in the energy matrix, the formation of black shales has a major impact on the geological carbon cycle. During photosynthesis, carbon dioxide in the atmosphere is captured to form organic matter; the burial of organic carbon in the sediments is thus a sink for $CO_2$. Gross primary productivity sequesters 120 billion metric tons of carbon a year on land and 92.2 billion metric tons a year in the ocean [2]. Increased primary production decreases atmospheric $CO_2$ (a known greenhouse gas), and therefore carbon burial may impact climate by decreasing the greenhouse effect.

The ability to find new (or expand known) petroleum plays is tied to our ability to recognize areas and/or stratigraphic intervals in which the depositional conditions were favorable to the accumulation of organics in the sediments. Likewise, the employment of approaches like ocean fertilization for the mitigation of $CO_2$ buildup in the atmosphere and global warming requires understanding of the controls on organic carbon burial to improve the efficiency of the method and account for potential unintended consequences.

The debate on whether high rates of primary productivity (leading to increased organic flux) or bottom-water anoxia (leading to preferential preservation of organic carbon) is more important in the accumulation of organics in the sediments has been ongoing for decades (e.g., [3–10], to cite a few). Clearly, primary productivity is a key factor; if the organic flux is low, the result will be organic-poor sediments (Figure 1). But anoxic conditions may be either a consequence of high primary productivity (as oxygen is consumed during microbial respiration) or a result of basin configuration. High primary productivity requires sustained nutrient input, which may be supplied by upwelling and/or riverine discharge. The discharge of fresh water to the ocean may lead to water stratification and stagnation, which may favor anoxic bottom conditions (Figure 1). Therefore, the organic content in mudrocks is the result of a complex system with multiple controls rather than a binary system. As such, teasing out the relative contribution of each control (e.g., primary productivity, redox conditions) to the outcome (i.e., organic content) is challenging.

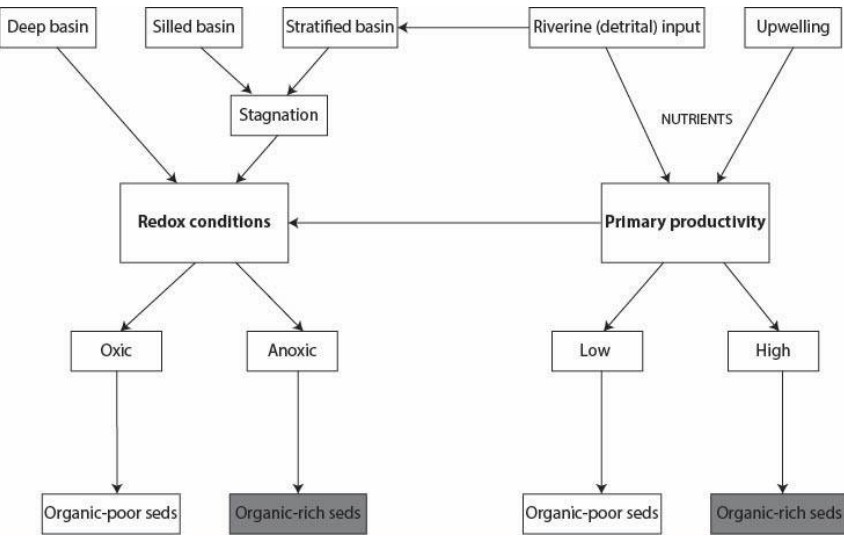

**Figure 1.** Simplified conceptual model of the controls on the burial of organics in the sediments.

Research on black shales oftentimes relies on the use of chemical indices that serve as proxies for the physical–chemical conditions of ancient bodies of water (e.g., [11–13]). Different chemical indices are used as proxies for different parameters, including redox conditions [14–16], salinity [17] and hydrographic conditions [18]. The reliability and limitations of these indices as indicators of the depositional conditions have been considered in previous publications [19–21] and hence are not being questioned here. In this work, we assume that the information provided by chemical proxies helps to assess the depositional conditions in the body of water, implicitly assuming that the chemical indices preserve the depositional signal. The challenge is then how to make sense of the (sometimes conflicting) information provided by chemical indices to determine the cause–effect relationship between depositional parameters, and to assign relative weights to possible drivers of the organic content in fine-grained sediments.

One common way of addressing this problem is to find correlations between two or more parameters. For instance, Fe/Al values greater than 0.6 are interpreted as indicative of anoxic bottom waters [22]; a positive correlation between Fe/Al and total organic carbon (TOC) could be taken as an indication that redox conditions control TOC. However, correlation does not imply causation. For instance, anoxia may result from high primary productivity, in which case it would not be the cause but an effect of high primary productivity (the latter which would be the primary driver of both anoxia and the high organic content). As such, the positive correlation would reflect a (hidden) driver that affects both variables. Correlation and causation are often confused because of our tendency to find explanations for seemingly related events. When two variables appear closely associated, we often assume that one is dependent on the other, implying a causal relationship (mean-

ing, one would be the result of the other), which is not necessarily true. When analyzing how variable y varies with changes in x, we must consider (i) whether (and how) y is affected by factors other than x, (ii) what the function connecting these two variables is, and (iii) how we can distinguish cause of x on y from a mere correlation [23].

Like the Earth systems, the field of Economics deals with many components which may interact with each other, forming complex systems. Economic theory states we should be suspicious of correlations found in observational data because they almost certainly do not reflect a causal relationship; variables are endogenously chosen by people who were making decisions they thought were best [23]. So how can one properly describe relationships and find causal links between variables interacting within a complex system? One way is to use statistical methods and models. In Economics, they are vital to understand data, predict trends, analyze economic theories, establish correlations between variables and assess potential causality.

The same can be applied to analyzing geological data. For the problem at hand (understanding what controls the accumulation of organics in the sediments), we use chemical indices as proxies for physical–chemical parameters. The overarching goal is to determine depositional conditions conducive of enrichment in organic matter, discriminating between different scenarios in terms of redox conditions, primary productivity, and method of delivery of nutrients, and to establish, in the process, the interdependence between and the relative importance of the different parameters to organic accumulation; in other words, which parameters (represented by chemical indices) are good predictors of TOC.

The interpretation of chemical indices (and the depositional conditions they are proxies for) is not straightforward. The integration of multiple indices in large datasets is complicated and laborious; indices indicative of the same parameter (e.g., Ti/Al and Zr/Al as indicators of detrital input) may not behave in the same fashion; some indices respond to more than one factor (e.g., Si/Al may be driven by detrital input and/or primary productivity; [24]). Lastly (and perhaps more importantly), multiple factors may impact the production and preservation of organics. The amount of organics (expressed as TOC) is the desired outcome of a chain of causal effects in a complex system, in which an unknown number of unobserved variables may interact.

The objective of this work is thus to explore how statistical methods and models can be applied to process geological data, with the goal of helping to establish causal links between variables, detect unobservable variables and assign the relative importance of multiple variables in a complex system. The end goal is to understand how the interconnectedness of Earth systems operates, identifying controlling mechanisms to resolve geological issues. For this exercise, the chosen issue was what controls the accumulation of organic matter in the mudrocks of the Woodford/Chattanooga Formation. The analyzed variables were riverine (detrital) input and upwelling (as sources of nutrients for primary productivity), redox conditions and primary productivity (Figure 1).

## 2. Materials and Methods

The chemical indices selected for this analysis were Ti/Al, Zr/Al, Si/Al, Cu/Al, Ni/Al, Fe/Al, Mo/Al, U/Al, P/Al and U/Mo. Ti/Al and Zr/Al are proxies for riverine (detrital) input; Cu/Al and Ni/Al are proxies for primary productivity; Fe/Al and U/Mo are proxies for redox conditions. The other indices may be driven by more than one factor: Si/Al by riverine (detrital) input and/or primary productivity, Mo/Al and U/Al by redox conditions and/or primary productivity, and P/Al by upwelling and/or primary productivity.

The dataset on the Woodford/Chattanooga Formation included chemical indices calculated from the elemental concentrations collected with a hand-held X-ray fluorescence equipment in samples from a drill core in Kansas [25] and outcrop sections in Oklahoma [26], Alabama and Tennessee [27]. The dataset, available in the Supplementary Materials (Table S1), consists of a total of 38 datapoints, with ten samples per location (except for Oklahoma, with only eight datapoints). Data from the literature indicates that the Woodford/Chattanooga Formation is immature to early mature in the study areas, with $T_{max}$ ranging from 363 to 466 °C and Ro from 0.49 to 0.76 [28–32].

A series of statistical tests and models were performed in R. Some tests either yielded no relevant results (e.g., principal component analysis, in which 95% of the variations was explained by the first principal component) or misleading results (e.g., linear regression with imputation), and therefore were not pursued any further.

The analyses deemed useful for the studied issue are cluster analysis, discrimination and classification analysis and linear (=ordinary least squares) regression.

### 2.1. Cluster Analysis

Cluster analysis, or clustering, is a statistical technique that aims to group similar items into clusters, or groups, based on their characteristics. Each cluster consists of items that are more similar to each other than they are to items in other clusters. This technique is used in many fields, including machine learning, data mining, pattern recognition, image analysis, information retrieval, and bioinformatics. There are several different clustering methods; however, we chose to use a partitional clustering method called k-means, in which the method attempts to split the data into k distinct non-overlapping clusters [33].

### 2.2. Discrimination and Classification Analysis

Discrimination and classification analysis refers to a set of techniques in machine learning and statistics that are used to predict categorical outcomes. Specifically, we used linear discriminant analysis (LDA); the method assumes that the classes are linearly separable, and it finds the linear combination of features that best separates the classes. LDA also assumes that the predictor variables are normally distributed and that the classes have the same covariance matrix [34].

### 2.3. Linear Regression Analysis

Ordinary Least Squares (OLS) is a method used in linear regression to estimate the unknown parameters in a linear regression model. The goal is to minimize the sum of the squared residuals, hence the name "least squares". We posit a relationship between the dependent variable and the independent variables of the form $y = X\beta + \varepsilon$, where $y$ is a vector of the dependent variable observations, $X$ is a matrix of the observed values of the independent variables (each column corresponds to one variable), $\beta$ is a vector of the unknown parameters that need to be estimated, and $\varepsilon$ is a vector of error terms, representing the difference between the observed and predicted values of the dependent variable (the residuals). We use the OLS method to find the values of $\beta$ that minimize the sum of the squared residuals. The method of OLS makes several key assumptions. First, we assume the relationship between the independent and dependent variables is linear. Second, the observations are independent of each other. In the context of time series data, this assumption means that there is no correlation between the residuals (errors) of different time periods. Third, we assume homoscedasticity, in which the variance of the errors is constant across all levels of the independent variables. In other words, the spread of the residuals is the same for all predicted values. If this assumption is violated, we have a problem called heteroscedasticity. This can be corrected by using robust standard errors if necessary. Fourth, this method assumes normality, or that the errors (residuals) are normally distributed. If this assumption is violated, it can affect the hypothesis tests that we might want to run to check the significance of the coefficients. Lastly, no multicollinearity: the independent variables are not perfectly correlated with each other. This assumption is important because if two variables are highly correlated, it is hard to disentangle the effect of one from the effect of the other [35].

The OLS analysis for the Woodford/Chattanooga Formation was performed with the purpose of checking which chemical indices are significant to predict TOC. Therefore, TOC was the dependent variable, and the initial selection of the chemical indices was performed by using a stepwise selection process. In this process, the bidirectional algorithm starts with both the full model and an empty model, then chooses whether adding a variable or removing a variable would improve the goodness of fit the most. We used

the Akaike Information Criterion (AIC) as the measure of goodness of fit, since it tends to select larger models as compared to the Bayesian Information Criterion (BIC), which puts a higher penalty on having many variables. The algorithm keeps either adding or removing variables until no further action can improve the model, resulting in the best fit for the data. This process also helps to assure that there is no perfect collinearity between variables, as if two variables are perfectly collinear, they would explain the same variation in the data. Thus, removing one of them would be strictly improving the model.

*2.4. Sensitivity Test*

To test the method sensitivity, chemical data for modern environments with different redox conditions were compiled and chemical indices were calculated. The advantage of using data from modern environments is that the oxygen concentration in bottom waters can be measured and hence the redox conditions are established with certainty (rather than estimated from indirect methods such as chemical proxies). We compiled data on Al, Fe, Mo and U concentrations in modern environments with oxic (>60 μM $O_2$), hypoxic (4–40 μM $O_2$) and anoxic (<2 μM $O_2$) bottom waters and calculated Fe/Al, Mo/Al, U/Al and U/Mo. The dataset, available in the Supplementary Materials (Table S2), comprises 12 entries for oxic, 47 for hypoxic and 96 for anoxic environments. Chemical data to construct this database were extracted from measurements obtained along the Eastern Pacific in Peru [36] and in the Black Sea [20], the latter location providing only two data entries for anoxic environments.

Only chemical indices proxies for redox conditions were used in the sensitivity test because of the suitable superposition between (i) availability of chemical data to calculate the indices and (ii) the existence of detailed measurements of the oxygen concentration in the bottom waters, which provided a good control on the redox conditions. The other two parameters (detrital input and primary productivity) lacked either data on the elemental concentration to calculate desired indices or measurements of required parameters (detrital and organic fluxes). With the dataset on modern environments, we ran a cluster analysis, as well as a discrimination and classification analysis to test whether the chemical indices could predict redox conditions well.

Following the sensitivity test, we ran the same series of analyses on the Woodford/Chattanooga Formation dataset (38 data points), plus linear regression analysis. The cluster analysis included all the chemical indices (Ti/Al, Zr/Al, Si/Al, Cu/Al, Ni/Al, Fe/Al, Mo/Al, U/Al, P/Al, U/Mo) and TOC. The purpose was to check whether the chemical indices would automatically group into clusters related to TOC and/or with any environmental significance.

We ran a series of linear regression models, initially including only the individual chemical indices, and then adding possible interactions between certain indices to assess the relative interdependence of these parameters. The interactions added to the models were based on conceptual models of what controls the burial of organic matter in the sediments (e.g., [5,24]) (Figure 1). The modeled interactions were riverine input in primary productivity, upwelling in primary productivity, primary productivity in redox conditions, and finally all interactions altogether.

Additionally, to test the robustness of the interactions, we conducted transformations on the data used to see whether stabilizing the variance and standardizing the mean of the indexes would impact the analysis. The two transformations that yielded the most Gaussian-like distributions were the log transform (which is simply the logarithm of each index) and the Tukey transformation. The Tukey transformation is widely used in data analysis to help meet the assumptions of statistical tests or linear models. It can help stabilize variance, make the data more normally distributed, or linearize relationships between variables. It is especially valuable in exploratory data analysis, helping to reveal underlying patterns or relationships that are not apparent in untransformed data. It is not always appropriate to use, especially because the interpretation of the coefficients becomes much more challenging, since the scale of the data changes. In our case, however, we

were less concerned about the magnitude of the coefficients, but rather whether we should consider them at all; thus, the interpretation problem was diminished.

The Tukey transformation is defined as follows:

$y = (x^{\lambda-1})/\lambda$, if $\lambda \neq 0$;

$y = \log(x)$, if $\lambda = 0$.

Different values of $\lambda$ correspond to different types of transformations:

- $\lambda = -1$ is a reciprocal transformation.
- $\lambda = -0.5$ is a reciprocal square root transformation.
- $\lambda = 0$ is a log transformation.
- $\lambda = 0.5$ is a square root transformation.
- $\lambda = 1$ is no transformation (identity).
- $\lambda = 2$ is a square transformation.
- $\lambda = 3$ is a cube transformation.

We then used maximum likelihood estimation to find the $\lambda$ that maximizes the likelihood of the transformed data with a Gaussian (normal) distribution.

A comparison between the raw data distribution and the log and Tukey transformations for each chemical index is shown in Figure 2.

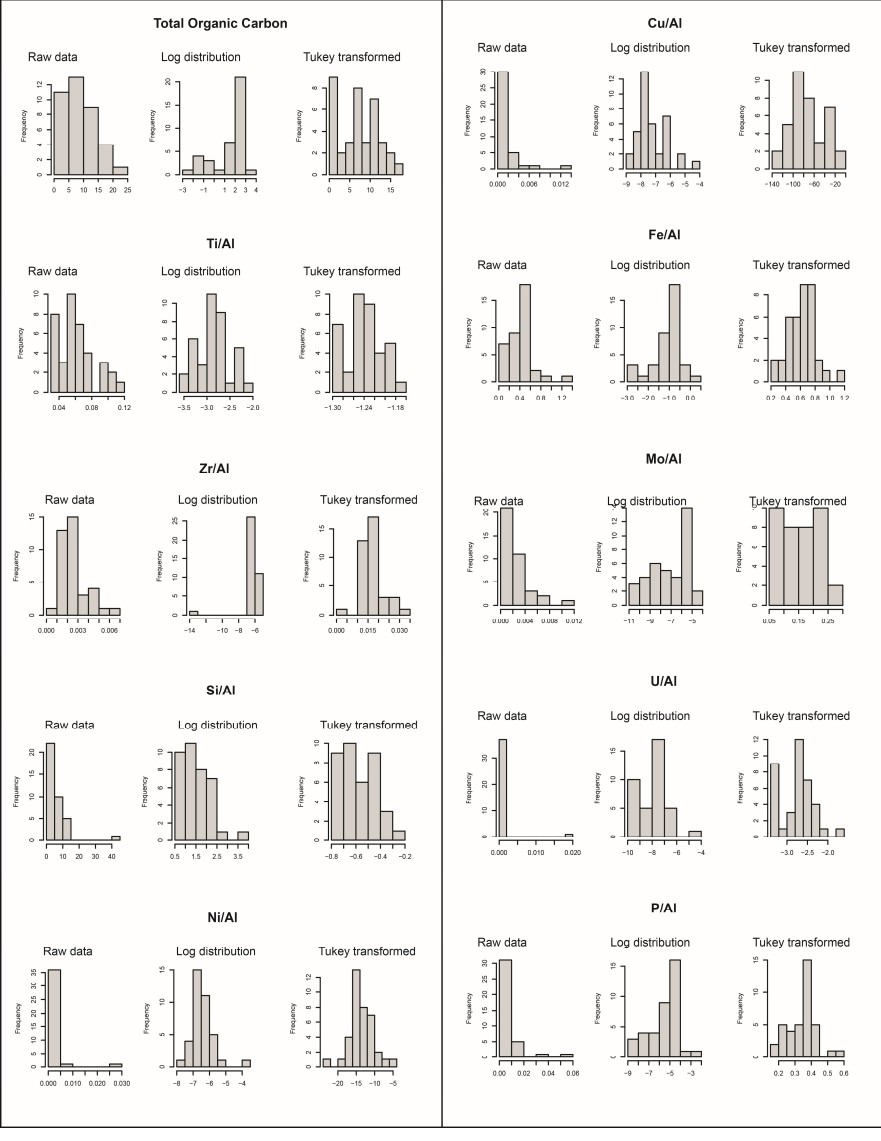

**Figure 2.** Data distribution for the different variables utilized in this study (total organic carbon plus the selected chemical indices). For each variable, the raw data, and the log and Tukey transformed distributions are shown.

## 3. Results and Discussion of Geological Significance

### 3.1. Sensitivity Test

Cluster analysis on modern environments was performed for two, three and four clusters, and using both the raw and the normalized data. Data normalization was carried out because the chemical indices have very different scales, with Fe/Al varying between zero and one, and Mo/Al and U/Al with values of $10^{-4}$ or $10^{-5}$, a three to four order of magnitude difference. Three clusters fit the data best, with the normalized data falling into three well-defined clusters (Figure 3). The values for the center points of each cluster are displayed in Table 1. Despite the better definition of the clusters with the normalized data, the values for the center points no longer have a direct geological meaning. The negative values for the normalized data result from the logarithm function used to normalize the variables. In log (1) = 0, the log of a number greater than one is positive and the log of a number between zero and one is negative. Many of the indices were between zero and one.

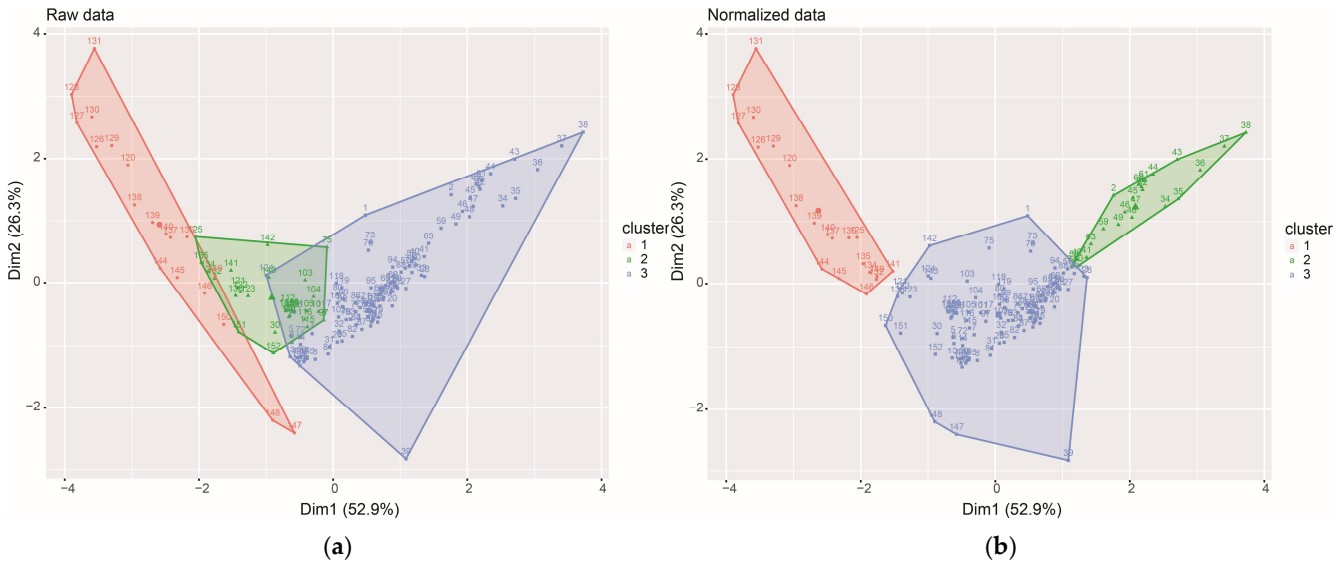

**Figure 3.** Cluster plots on redox conditions in modern environments with three clusters, using (**a**) raw data and (**b**) normalized data. Cluster 1 in red, 2 in green and 3 in blue.

**Table 1.** Chemical indices for the center point in each cluster of Figure 3, using the raw and the normalized data.

| Cluster | Fe/Al | Mo/Al | U/Al | U/Mo |
|---|---|---|---|---|
| 1 (raw) | 0.6359066 | $2.210526 \times 10^{-5}$ | $2.115789 \times 10^{-4}$ | 8.5842105 |
| 2 (raw) | 0.4594432 | $7.466667 \times 10^{-5}$ | $2.53 \times 10^{-4}$ | 3.7733333 |
| 3 (raw) | 0.4141516 | $8.419811 \times 10^{-4}$ | $3.530189 \times 10^{-4}$ | 0.5635849 |
| 1 (normalized) | 1.770597 | $-0.8740039$ | $-0.394565$ | 2.0753108 |
| 2 (normalized) | $-0.4868763$ | 1.4427397 | 1.8343059 | $-0.5652342$ |
| 3 (normalized) | $-0.2317273$ | $-0.1479243$ | $-0.3248862$ | $-0.2728719$ |

Sediments deposited in oxic conditions have Fe/Al less than 0.5 [19] and U/Mo greater than 1.5 [37]. U/Mo appears to be the best chemical index to differentiate oxic from anoxic conditions in modern environments, with clearly distinct and meaningful center-point values. Based on U/Mo, Cluster 1 represents oxic, Cluster 2 suboxic and Cluster 3 anoxic conditions. The data on modern environments shows a wide variation of Fe/Al under the same redox conditions, which results in values for the cluster center points around 0.5 (0.4–0.6). Therefore, despite the widespread use of Fe/Al as a paleoredox indicator in the geological literature, this analysis suggests that this chemical index should be used in conjunction with other proxies for redox conditions, not taken at point value.

The discrimination and classification analysis on modern environments tested whether the chemical indices could predict redox conditions. The results with the raw and the normalized data were the same. Out of the 96 data points deposited in anoxic conditions, the chemical indices accurately predicted the redox condition as "anoxic" in 93 of them, a prediction rate of 97%. The chemical indices correctly predicted 36 out of 47 data entries for hypoxic conditions (a prediction rate of 77%), and 7 out of 12 data entries for oxic conditions (prediction rate of 58%). As expected, the prediction rate decreased because of smaller sample sizes. Data points for oxic conditions were almost an order of magnitude less than for anoxic conditions. Nonetheless, the analysis indicated that the selected chemical indices confidently discriminate between anoxic and oxic conditions.

### 3.2. Cluster Analysis of the Woodford/Chattanooga Formation

The cluster analysis on the Woodford/Chattanooga dataset was performed for two, three and four clusters using the raw data, as well as log distribution and Tukey transformed data, considering all chemical indices and total organic carbon (TOC). For the raw data, three clusters fit the data best, with one outlier (Figure 4a). For both the log distribution and the Tukey transformed, two clusters are best (Figures 4b and 4c, respectively), with two outliers in the log distribution data (Figure 4b).

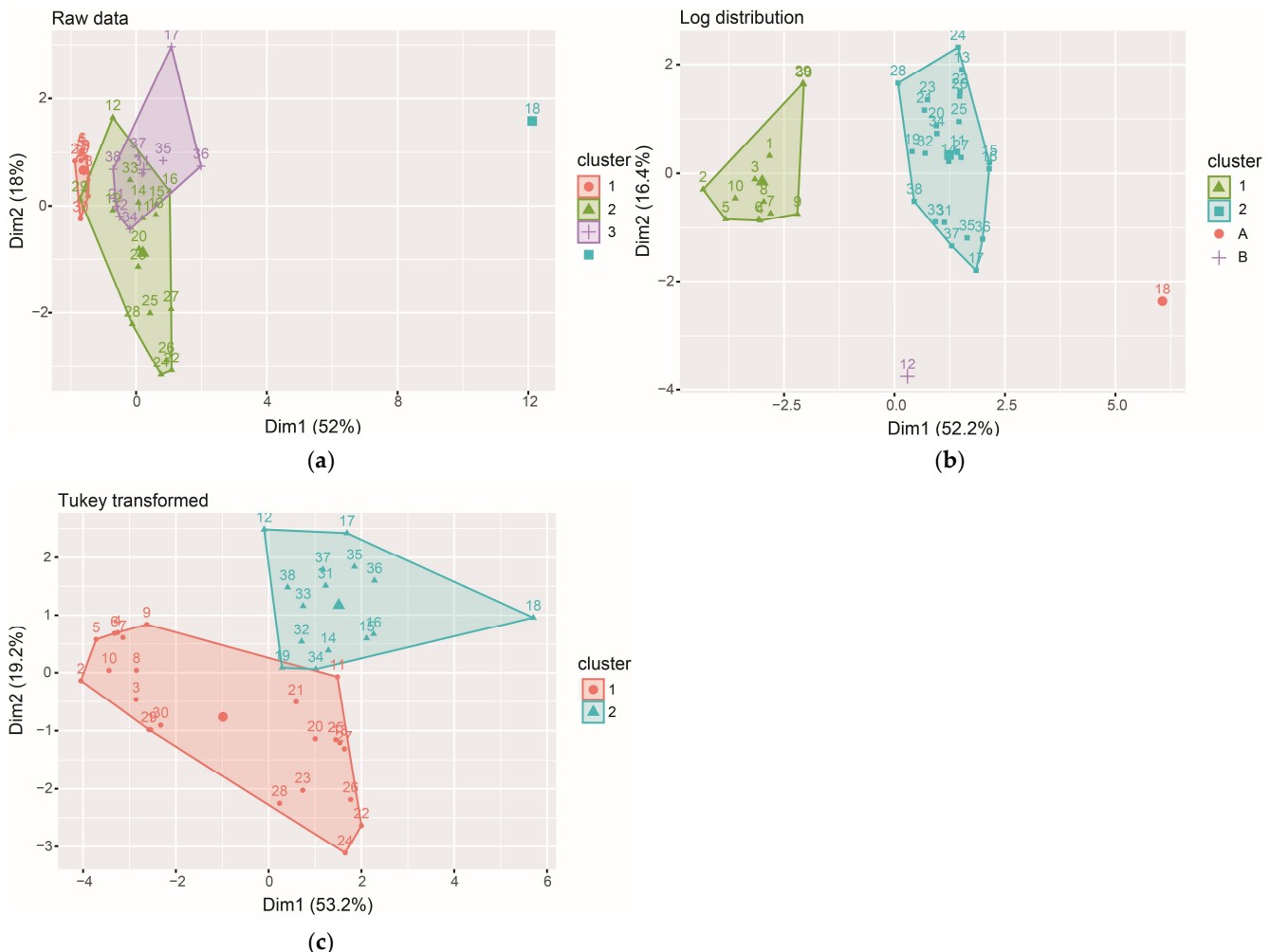

**Figure 4.** Cluster plots for the Woodford/Chattanooga Formation dataset, including total organic carbon (TOC) and all chemical indices, using (**a**) raw data (with outlier in blue), (**b**) log distribution and (**c**) Tukey transformed data.

The values for the center points of each cluster are displayed in Table 2. The clusters are characterized by different TOC values. For the raw data, Cluster 1 displays the lowest TOC (1%) and Cluster 3 the highest TOC (16%); the outlier and Cluster 2 have intermediate TOC values (8–9%). The fact that the clusters display different mean TOC suggests that the chemical indices can predict TOC. The existence of an outlier in the raw data ensues from the very high Si/Al (42, as opposed to 2–8 in the three clusters). The center point values in the log and Tukey transformed distributions do not have a geological meaning, especially considering negative values for TOC and chemical indices. However, it is interesting to note that the two clusters in the Tukey transformed distribution have mean TOC values of 4 and 11%, also separating populations according to TOC.

**Table 2.** Chemical indices for the center point in each cluster (and outliers) of Figure 4, using the raw, log and Tukey transformed data.

| Cluster | TOC | Ti/Al | Zr/Al | Si/Al | Ni/Al | Cu/Al | Fe/Al | Mo/Al | U/Al | P/Al |
|---------|-----|-------|-------|-------|-------|-------|-------|-------|------|------|
| 1 (raw) | 1.08 | 0.039 | 0.0017 | 2.1356 | 0.0013 | 0.0004 | 0.4009 | 0.0001 | $8.6 \times 10^{-5}$ | 0.0011 |
| 2 (raw) | 8.92 | 0.077 | 0.003 | 7.685 | 0.002 | 0.0007 | 0.3864 | 0.0025 | $5.34 \times 10^{-4}$ | 0.0083 |
| 3 (raw) | 15.67 | 0.058 | 0.0021 | 4.2566 | 0.0021 | 0.0033 | 0.3844 | 0.0028 | $8.24 \times 10^{-4}$ | 0.0093 |
| Outlier | 7.58 | 0.069 | 0.0042 | 42.3083 | 0.0256 | 0.0065 | 1.29 | 0.0116 | $1.81 \times 10^{-2}$ | 0.0552 |
| 1 (log) | −0.39 | −3.237 | −6.376 | 0.7695 | −6.9783 | −7.7345 | −1.1613 | −9.2254 | −9.359 | −7.15589 |
| 2 (log) | 2.39 | −2.663 | −5.9383 | 1.7988 | −6.2902 | −7.0771 | −1.1184 | −6.154 | −7.4749 | −4.8239 |
| Outlier A | 2.03 | −2.665 | −5.4684 | 3.745 | −3.6643 | −5.0391 | 0.2546 | −4.4569 | −4.011 | −2.896 |
| Outlier B | 1.95 | −3.056 | −13.5158 | 1.636 | −5.9441 | −6.484 | −0.6733 | −7.2207 | −7.6158 | −4.9042 |
| 1 (Tuk.) | 4.44 | −1.24 | 0.0178 | −0.61420 | −14.8827 | −93.987 | 0.6169 | 0.124 | −2.9252 | 0.3146 |
| 2 (Tuk.) | 11.65 | −1.237 | 0.0152 | −0.5382 | −11.4773 | −39.0194 | 0.6467 | 0.206 | −2.4555 | 0.3859 |

Comparing Clusters 1 and 3 (with the lowest and highest mean TOC, respectively), the main difference is the mean values for Si/Al, Cu/Al, Mo/Al, U/Al and P/Al. The cluster with the highest TOC (Cluster 3) has Si/Al twice as high as Cluster 1, and an order of magnitude higher for the other chemical indices. Since Si/Al and Cu/Al are proxies for primary productivity, Mo/Al and U/Al proxies for degree oxygenation (possibly driven by primary productivity), and P/Al proxy for upwelling (which may drive primary productivity), these results allude to the importance of primary productivity in controlling TOC.

*3.3. Linear Regression Analysis of the Woodford/Chattanooga Formation*

The linear regression analysis, with TOC as the dependent variable, included all the chemical indices. The purpose of this analysis was to check which indices are significant to predict TOC. A series of models were run using the raw data, as well as log and Tukey transformed data. In all the models, TOC was the dependent variable, and all chemical indices were included, but in some we added interactions between chemical indices to check whether they were significant (and thus interdependent), evaluating the effect of including interactions between proxies for certain depositional parameters in the model outcome. These possible interactions were based on the conceptual model of what controls TOC (Figure 1); e.g., riverine delivery of nutrients affecting primary productivity, or primary productivity-driven bottom-water anoxia.

For the different data distributions, we ran models with all the individual chemical indices (i) with no interaction between chemical indices proxies for depositional conditions, (ii) with the interaction between proxies for detrital input and primary productivity, (iii) with the interaction between proxies for upwelling and primary productivity, (iv) with the interaction between proxies for redox and primary productivity, and (v) with the interaction between proxies for detrital input, redox conditions, primary productivity and upwelling. The results of each model are summarized below (Table 3). The full results, including evaluation metrics such as F statistics and *p* values, are available in Supplementary Materials (Table S3).

**Table 3.** Summary of the results from the different models in the linear regression analysis, using the raw, log and Tukey transformed data. Most relevant (statistically significant) chemical indices and interactions to predict TOC by themselves in each model are in italics.

| Model | Raw Data | Log Distribution | Tukey Transformed |
|---|---|---|---|
| (i) No interaction | $R^2$ 0.7484 (adjusted $R^2$ 0.6675); relevant indices Ti/Al, Mo/Al, Si/Al, Cu/Al, Zr/Al | $R^2$ 0.8396 (adjusted $R^2$ 0.7881); relevant indices Ti/Al, Mo/Al, Fe/Al, (Si/Al, Cu/Al) | $R^2$ 0.7984 (adjusted $R^2$ 0.7336); relevant indices Ti/Al, Mo/Al, Fe/Al, (Si/Al, Cu/Al) |
| (ii) Interaction between detrital input and primary productivity | $R^2$ 0.801 (adjusted $R^2$ 0.7054); relevant indices Ti/Al, Mo/Al; significant interaction between Ti/Al and Si/Al | $R^2$ 0.8835 (adjusted $R^2$ 0.8275); relevant indices Cu/Al, U/Al, Mo/Al; significant interaction between Ti/Al and Cu/Al | $R^2$ 0.8415 (adjusted $R^2$ 0.7654); relevant indices Si/Al, Mo/Al; significant interaction between Ti/Al and Si/Al |
| (iii) Interaction between upwelling and primary productivity | $R^2$ 0.8644 (adjusted $R^2$ 0.7909); relevant indices U/Al, Mo/Al; no significant interaction between indices | $R^2$ 0.8837 (adjusted $R^2$ 0.8208); relevant indices U/Al, (P/Al, Cu/Al); significant interaction between P/Al and Cu/Al | $R^2$ 0.8702 (adjusted $R^2$ 0.7999); relevant indices U/Al, (Si/Al); significant interaction between Si/Al and P/Al |
| (iv) Interaction between redox conditions and primary productivity | $R^2$ 0.9304 (adjusted $R^2$ 0.8712); relevant indices U/Al, Cu/Al, (Ni/Al) significant interactions between Cu/Al and Mo/Al, U/Al and Mo/Al, Ni/Al and Fe/Al, Ni/Al and U/Al, Cu/Al and Fe/Al, Cu/Al and U/Al, Mo/Al and Fe/Al, (Ni/Al and Mo/Al) | $R^2$ 0.9258 (adjusted $R^2$ 0.8628); relevant index U/Al; no significant interaction between indices | $R^2$ 0.895 (adjusted $R^2$ 0.8058); no relevant indices; significant interaction between Ni/Al and U/Al |
| (v) All interactions (between detrital input, redox conditions, primary productivity and upwelling) | $R^2$ 0.9586 (adjusted $R^2$ 0.8822) relevant indices Ni/Al, Cu/Al and Zr/Al significant interactions between Cu/Al and Mo/Al, Cu/Al and U/Al, U/Al and Mo/Al, Cu/Al and Fe/Al, Ti/Al and Ni/Al | $R^2$ 0.9456 (adjusted $R^2$ 0.8451); no relevant indices; no significant interaction between indices | $R^2$ 0.9338 (adjusted $R^2$ 0.8114); no relevant indices; significant interactions between Ti/Al and Ni/Al, Ti/Al and Cu/Al, Si/Al and P/Al, Ni/Al and Fe/Al, Mo/Al and Fe/Al |

The results show that even the simple models (i), where <u>no interactions</u> are considered, fit the data well, with an $R^2$ of 0.7484 for the raw data, 0.8396 for the log distribution and 0.7984 for the Tukey transformed data. Statistically significant chemical indices to predict TOC by themselves in these models are, in decreasing order of relevance, Ti/Al, Mo/Al, Si/Al and Cu/Al. Zr/Al was statistically significant using the raw data and Fe/Al using the log distribution and Tukey transformed data.

Adding the <u>interaction between detrital input and primary productivity</u> to the models (ii) improves the results, with an $R^2$ of 0.801 for the raw data, 0.8835 for the log distribution and 0. 8415 for the Tukey transformed data. Assuming that the interaction between detrital input and primary productivity is the driver of TOC, statistically significant chemical indices to predict TOC by themselves are Ti/Al, Mo/Al (raw data), Cu/Al, U/Al and Mo/Al (log distribution) and Si/Al, Mo/Al (Tukey transformed). Considering interactions between chemical indices, the interactions between Ti/Al and Si/Al (raw data and Tukey transformed), and Ti/Al and Cu/Al (log distribution) are statistically significant. Interactions that are not "statistically significant" are those which, if added to the model, do not help predict TOC any better than without including them, simply adding noise to the model.

If the <u>interaction between upwelling and primary productivity</u> (iii) is added to the models instead, they fit the data even better, with an $R^2$ of 0.8644 for the raw data, 0.8837 for the log distribution and 0.8702 for the Tukey transformed data. Assuming that the interaction between upwelling and primary productivity is the driver of TOC, the most statistically significant chemical index to predict TOC by itself is U/Al, followed by Mo/Al.

Even though this interaction is irrelevant when the raw data are used, the model ran with the log distribution and the Tukey transformed data show that interactions between P/Al and Cu/Al (log distribution) and P/Al and Si/Al (Tukey transformed) are statistically significant (which means that adding them to the models improves the outcome).

The models that included the interaction between redox conditions and primary productivity (iv) fit the data very well, with $R^2$ of 0.9304 for the raw data, 0.9258 for the log distribution and 0.895 for the Tukey transformed data. The fact that more than 90% of the data is explained by these models is promising, suggesting that the assumptions regarding the controls on the burial of organic matter in the sediments are appropriate; specifically, it suggests that the productivity-driven anoxia exerts substantial control on TOC. The use of raw data yielded much better results than of log distribution or Tukey transformed data. The latter has also not been helpful to indicate statistically significant chemical indices or interactions between indices for prediction of TOC. With the raw data, however, if the interaction between redox conditions and primary productivity drives TOC, statistically significant chemical indices to predict TOC by themselves are U/Al and Cu/Al. Regarding interactions between chemical indices, statistically significant interactions are found between Cu/Al and Mo/Al, U/Al and Mo/Al, Ni/Al and Fe/Al, Ni/Al and U/Al, Cu/Al and Fe/Al, Cu/Al and U/Al, Mo/Al and Fe/Al.

Finally, the models that included all the interactions (between detrital input, redox conditions, primary productivity, and upwelling) (v) yielded the best results, with the models explaining about 95% of the data. Once again, the best $R^2$ (0.9586) were achieved using raw data, in comparison with both log distribution ($R^2$ = 0.9456) and Tukey transformed data ($R^2$ = 0.9338). The excellent fit indicates that all these interactions together predict TOC better than without them, which in geological terms can be translated as "all the factors depicted in Figure 1 as controlling the amount of organics buried in the sediments are relevant and the conceptual model is good, albeit complex". With the raw data, considering that all these interactions drive TOC, statistically significant chemical indices to predict TOC by themselves are Ni/Al, Cu/Al and Zr/Al. These suggest that primary productivity (represented by Ni/Al and Cu/Al) and detrital input (represented by Zr/Al) have a significant role in the organic burial. Statistically significant interactions between chemical indices are found between Cu/Al and Mo/Al, Cu/Al and U/Al, U/Al and Mo/Al, Cu/Al and Fe/Al, Ti/Al and Ni/Al. In geological terms, these might suggest the importance of primary productivity blooms (driven by riverine nutrient delivery), resulting in bottom-water anoxia.

The all-interactions model (v) included 24 variables and interactions. The post-stepwise all-interactions model using raw data removed one index (Si/Al) and four interactions (Ti/Al and Si/Al, P/Al and Si/Al, P/Al and Cu/Al, P/Al and Mo/Al). The implication of this action by the algorithm is that these five indices and interactions are not statistically significant to improve the model. Using the log distribution data, the algorithm removed the six interactions (P/Al and Cu/Al, Ni/Al and Mo/Al, Cu/Al and Fe/Al, Cu/Al and U/Al, Mo/Al and Fe/Al, Mo/Al and U/Al), maintaining 18 variables and interactions in the model. Using the Tukey transformed data, the post-stepwise all-interactions model removed four interactions (P/Al and Ni/Al, P/Al and Cu/Al, Cu/Al and Fe/Al, Cu/Al and Mo/Al), resulting in a total of 20 variables and interactions. Statistical parameters comparing the simple model (i), full model with all interactions (v) and the post-stepwise model using raw, log and Tukey transformed data distributions are included in Supplementary Materials (Table S4) and summarized in Table 4.

Only the interaction between P/Al and Cu/Al was removed in the post-stepwise model using all three data distributions (raw, log and Tukey transformed), which could be interpreted as upwelling (represented by P/Al) not strongly impacting primary productivity (represented by Cu/Al). Removal of the interaction between P/Al and some indices (e.g., Mo/Al, Ni/Al) using the other data distributions seem to corroborate that upwelling does not appear to be a strong driver of primary productivity and anoxia to control TOC in this dataset.

Removal of Si/Al and its interactions using the raw data might reflect the difficulty in establishing the main driver of this index, since both primary productivity and detrital input affect Si/Al, which adds "noise" to the model.

**Table 4.** Comparison between the statistical parameters for the simple, full and post-stepwise models using raw, log and Tukey transformed data distributions.

| | Raw Data | | | Log Distribution | | | Tukey Transformed | | |
|---|---|---|---|---|---|---|---|---|---|
| | Simple Model (i) | Full Model (v) | Post-Stepwise Model | Simple Model (i) | Full Model (v) | Post-Stepwise Model | Simple Model (i) | Full Model (v) | Post-Stepwise Model |
| $R^2$ | 0.748 | 0.959 | 0.957 | 0.84 | 0.946 | 0.943 | 0.798 | 0.934 | 0.932 |
| $R^2$ Adj. | 0.668 | 0.882 | 0.911 | 0.788 | 0.845 | 0.888 | 0.734 | 0.811 | 0.851 |
| AIC | 210.1 | 171.5 | 163.3 | 91.9 | 80.8 | 70.8 | 190.5 | 178.2 | 171.4 |
| BIC | 228.1 | 214.1 | 197.7 | 109.9 | 123.4 | 103.6 | 208.5 | 220.8 | 207.4 |
| Log.Lik. | −94.057 | −59.758 | −60.667 | −34.956 | −14.421 | −15.424 | −84.263 | −63.117 | −63.684 |
| F | 9.254 | 12.55 | 20.88 | 16.288 | 9.412 | 17.344 | 12.319 | 7.635 | 11.603 |
| RMSE | 2.88 | 1.17 | 1.19 | 0.61 | 0.35 | 0.36 | 2.22 | 1.27 | 1.29 |

In any case, the selection of which indices/interactions are removed in the post-stepwise model using different data distribution stems from the model putting different weights depending on the data distribution. The key point, however, is that the models improve with added interactions, as shown by the increased $R^2$ from (i) to (v) (Table 3). The post-stepwise process applied to the all-interactions model enhances data fit even more, as evident from the analysis of residuals (Figure 5). The model with no interactions (i) shows an undesirable trend in the residuals (Figure 5a), whereas the post-stepwise selection, all-interactions model (v) displays no trend, only noise, in the residuals (Figure 5b).

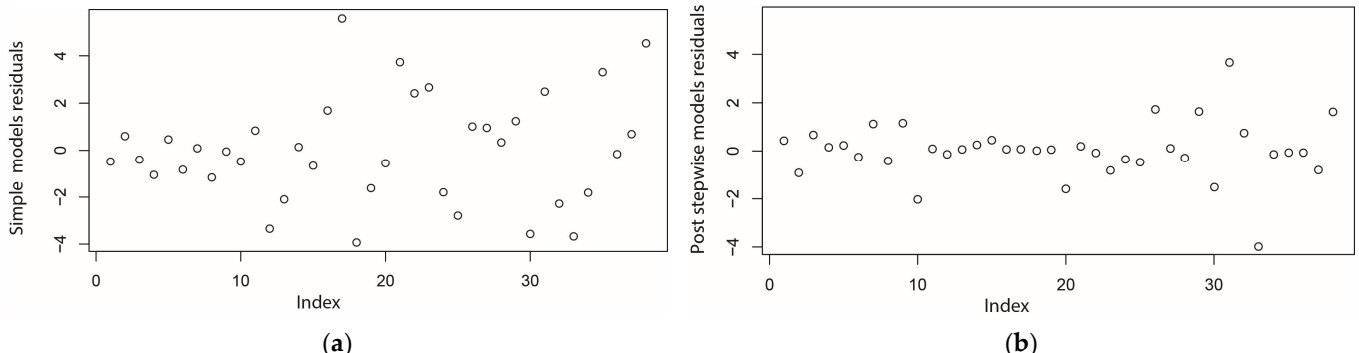

(**a**)            (**b**)

**Figure 5.** Residuals for the model with (**a**) no interactions (i), and (**b**) post stepwise, all interactions (v).

This exercise with linear regression models shows that (1) transforming the data (into log or Tukey transformed distribution) does not always improve the outcome (in other words, transformation does not always make the model fit the data better), (2) the "all-interactions" models (v) offer the best fit to the Woodford/Chattanooga dataset, regardless of data distributions, (3) Si/Al is not a good proxy for detrital input and should be avoided, especially in successions with abundant siliceous organisms where this index reflects the superposition of detrital input and primary productivity, (4) upwelling appears to be less significant than detrital input in controlling primary productivity and TOC, and (5) anoxia does not appear to be a primary driver of TOC.

## 4. Conclusions and Next Steps

Cluster analysis and ordinary least-squares (OSL) regression analysis proved to be valuable tools to assess the depositional parameters controlling the organic content in the sediments. The chemical indices calculated for the Woodford/Chattanooga Formation were

grouped in three clusters characterized by different TOC values, with the center points corresponding to 1%, 8–9% and 16% TOC. The fact that the clusters display different mean TOC suggests that the chemical indices can predict TOC. Other than TOC, the clusters are characterized by variations in the mean values for Si/Al, Cu/Al, Mo/Al, U/Al and P/Al, which signal the importance of primary productivity in controlling TOC.

The OSL model that includes "all-interactions" offers the best fit to the Woodford/Chattanooga dataset, explaining 95% of the data. This indicates that detrital input, primary productivity and bottom-water anoxia are relevant drivers of the amount of organics buried in the sediments, verifying the complexity of the system. However, OSL results also suggest that primary productivity and detrital input might have a more significant role in organic burial, whereas bottom-water anoxia might affect a secondary control, resulting from high primary productivity (triggered dominantly by detrital input).

A future development would be to ascertain the relative weight of each factor controlling TOC. We will attempt to test each individual pathways depicted in Figure 1 on the relative contribution to TOC through the application of two-step least squares (2SLS). In the forthcoming exercise, we intend to, for example, use the chemical index proxy for upwelling index (P/Al) to predict primary productivity (instead of using productivity proxies directly), isolating this first segment of the pathway between upwelling and TOC. Then, in a second stage, we will use the "predicted information" (instead of real chemical index proxies for primary productivity) to predict TOC. If this two-stage process predicts TOC well, the isolated pathway (upwelling primary productivity TOC) is strong and thus is an important channel controlling TOC. This will be performed for the different parts of the conceptual model in Figure 1 to assign the relative importance of each pathway.

Despite the valuable insights provided by the methods applied in this study, they only assist in the interpretation of complex Earth systems. It is still the geoscientist's role to make sense of what the results mean to sensibly interpret the depositional conditions and their implications for the accumulation of organic matter in the sediments.

**Supplementary Materials:** The following supporting information can be downloaded at: https://www.mdpi.com/article/10.3390/geosciences14020043/s1, Table S1: Woodford indices; Table S2: Modern redox; Table S3: OSL results; Table S4 Model comparison.

**Author Contributions:** Conceptualization; methodology; validation; formal analysis; writing—original draft preparation and review and editing; visualization, K.G. and L.G.D.R.; software, L.G.D.R.; supervision; project administration; funding acquisition, K.G. All authors have read and agreed to the published version of the manuscript.

**Funding:** This research was supported by the donors of the American Chemical Society Petroleum Research Fund under Undergraduate Research Grant "PRF # 59798-UR8".

**Data Availability Statement:** The raw data compiled for this study are available on request from the corresponding author.

**Conflicts of Interest:** The authors declare no conflict of interest.

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
