# Peer review of "Applying Statistical Analysis and Economics Models to Unscramble the Depositional Signals from Chemical Proxies in Black Shales"

_geosciences, doi:10.3390/geosciences14020043_

Round 1

Reviewer 1 Report

Comments and Suggestions for Authors

The authors have examined bulk geochemical data for samples from the Woodford Shale using a suite of statistical tools often used for econometric studies in order to discern the relative importance of primary productivity, preservation (redox) conditions, and detrital dilution/nutrient input on organic matter enrichment. The results of the study indicate that organic enrichment in the Woodford Shale is more influenced by productivity and detrital input than redox conditions.

Overall the paper is well written and demonstrates the utility of the statistical tools the authors have tested. That said, there are a number of studies that employ similar approaches or apply chemometric tools to source rock data in order to infer relationships between different rock properties. I have a few technical concerns (summarized below) but otherwise see no reason the paper shouldn’t be published in Geosciences after minor revisions.

One key factor that needs to be mentioned in the paper is the thermal maturity of the samples that were used, preferably in the form of maceral reflectance (vitrinite or solid bitumen) or even Tmax from programmed pyrolysis (though this should be checked against other published Woodford studies). If these source rocks mature for hydrocarbon generation then the TOC values will not represent the OM through diagenesis, and so the relationships to elemental ratios will be altered.

Another issue is the limited amount of data the authors have utilized. I don’t think they need to have collected additional new data, but there are many published chemostratigraphic and formation-wide geochemical studies of source rock units around the world that could have been evaluated as part of this work. I realize this paper represents something of a proof of concept, but including a similar analysis applied to a source rock with a different general lithofacies would demonstrate the robustness of the approach. The Stanford Sedimentary Geochemistry and Paleoenvironments project has a large collection of sample data that includes TOC and elemental concentrations, and the USGS has data available through ScienceBase.gov and the USGS Core Research Center website for a wide range of formations including the Bakken, Eagle Ford, Green River, Marcellus, Mowry, and Niobrara shales that could be examined.

Finally, there is no discussion of organic matter quality in this work (hydrogen content which impacts how oil-prone or gas-prone a source rock is). I realize is not the focus but like thermal maturity is a very important feature of source rock organic matter and is impacted by many of the same productivity-preservation-dilution processes as total organic richness.

Other comments

Page 1, Line 33: where did you find this definition of black shales? I would say this is more like a definition for source rocks generally, not all of which are commonly called black shales. Also, the TOC cutoff you note seems low to me – 2% is a more common cutoff in my experience.

Page 4, lines 131-136: why did you use values ratioed to aluminum content rather than enrichment factors? This normalizes the element/Al ratio to the same ratio for a standard shale (PAAS, NASC) or to crustal abundances. See Tribovillard et al. (cited in the manuscript) for details.

Table 1: there seems to be an issue with significant figures here.

The axes on Figures 3 and 4 are barely readable.

Line 470: change “not always improves” to “does not always improve”

Line 471: insert “transformation does” before “not”

Line 473: it may be worth mentioning here how biogenic silica throws this off.

Author Response

We are grateful for your comments and suggestions and we have changed the manuscript accordingly (highlighted in yellow in the revised manuscript). We have made all the modifications related to wording and grammar and tried to address all the issues pointed out. Please find the response to each point below. Your original comment is followed by our response in red.

One key factor that needs to be mentioned in the paper is the thermal maturity of the samples that were used, preferably in the form of maceral reflectance (vitrinite or solid bitumen) or even Tmax from programmed pyrolysis (though this should be checked against other published Woodford studies). If these source rocks mature for hydrocarbon generation then the TOC values will not represent the OM through diagenesis, and so the relationships to elemental ratios will be altered.

We do not have vitrinite reflectance or pyrolysis data for our samples specifically, but data available from the literature indicates that the Chattanooga/Woodford is mostly immature (to early mature) in the study areas (Lambert, 1992; Lambert 1993; Romero & Philp, 2012; Marlow, 2014, Lu 2015).

We added the sentence “Data from the literature indicates that the Woodford/Chattanooga Formation is immature to early mature in the study areas, with Tmax ranging from 363-466 °C and Ro from 0.49-0.76 [28-32]“ (lines 143-144 of the revised version).

Another issue is the limited amount of data the authors have utilized. I don’t think they need to have collected additional new data, but there are many published chemostratigraphic and formation-wide geochemical studies of source rock units around the world that could have been evaluated as part of this work. I realize this paper represents something of a proof of concept, but including a similar analysis applied to a source rock with a different general lithofacies would demonstrate the robustness of the approach. The Stanford Sedimentary Geochemistry and Paleoenvironments project has a large collection of sample data that includes TOC and elemental concentrations, and the USGS has data available through ScienceBase.gov and the USGS Core Research Center website for a wide range of formations including the Bakken, Eagle Ford, Green River, Marcellus, Mowry, and Niobrara shales that could be examined.

We agree that applying the same technique to larger datasets would be ideal. However, the approach we employed requires the availability of concentration data for ALL the elements required to calculate the indices plus TOC. Even in large databases, usually there is one or another elemental concentration missing. For example, a search in the Stanford SGP’s dataset for the Woodford-Chattanooga did not retrieve all data required for our analysis.

We have not investigated other datasets because it would require some data mining and processing, which would delay the publication of our results.

Finally, there is no discussion of organic matter quality in this work (hydrogen content which impacts how oil-prone or gas-prone a source rock is). I realize is not the focus but like thermal maturity is a very important feature of source rock organic matter and is impacted by many of the same productivity-preservation-dilution processes as total organic richness.

We do not have pyrolysis data for our samples, and as you stated, this was not the focus of our analysis. In any case, Lambert (1992) reports that, for samples obtained in KS and OK, the hydrogen index does not decrease with Tmax, suggesting that kerogen composition has not been influenced by the latter.

Page 1, Line 33: where did you find this definition of black shales? I would say this is more like a definition for source rocks generally, not all of which are commonly called black shales. Also, the TOC cutoff you note seems low to me – 2% is a more common cutoff in my experience.

The cut-off value to define organic-rich (black) shales is variable, depending on the author, going as low as 0.5% TOC to 2%, as discussed by Arthur & Sageman (1994). The > 1%TOC we allude to in the manuscript is the definition suggested by Arthur & Sageman (1994). We have added this reference in line 33 to clarify.

Page 4, lines 131-136: why did you use values ratioed to aluminum content rather than enrichment factors? This normalizes the element/Al ratio to the same ratio for a standard shale (PAAS, NASC) or to crustal abundances. See Tribovillard et al. (cited in the manuscript) for details

Both element/Al ratios and enrichment factors are used routinely to assess the paleodepositional conditions. We chose to use only the element/Al ratios (not EFs) because both (element/Al and EFs) are calculated using the same data (i.e., elemental concentrations) and we wanted to avoid redundancies in the data input for the statistical analysis. The choice of using element/Al ratios (rather than EF) aimed at maintaining an internal coherence in the data input, meaning, using data (for the model) obtained through the same methods and in the same unit.

Table 1: there seems to be an issue with significant figures here.

We apologize but we do not understand what you mean by that.

The axes on Figures 3 and 4 are barely readable.

We modified the figures, increasing the fonts.

Line 470: change “not always improves” to “does not always improve”

OK (line 477)

Line 471: insert “transformation does” before “not”

OK (line 478)

Line 473: it may be worth mentioning here how biogenic silica throws this off.

We added the phrase “where this index reflects the superposition of detrital input and primary productivity” (lines 481-482).

Reviewer 2 Report

Comments and Suggestions for Authors

1. The introduction of the article contains too much content; the author needs to reduce it.

2. Figure 1 is suggested for modification; the conceptual model schematic is prone to confuse readers.

3. The content of the experimental methods and samples requires logical organization. The author must add more subheadings to distinguish sample sources and experimental methods. For example, experimental methods include cluster analysis, Ordinary Least Squares, etc.

4. In the experimental methods, the description of the least squares method is overly detailed and lengthy; a more concise presentation is recommended.

5. Figure 3 to 5 needs to be redrawn by the author; the relevant coordinates are not clearly presented.

6. The discussion and experimental results of the article need to be separated. The author needs to reduce the research content and rewrite subheadings. The word count in Manuscript 3.2 is excessively long.

Author Response

We are grateful for your comments and suggestions and we have changed the manuscript accordingly (highlighted in yellow in the revised manuscript). We have made all the modifications related to wording and grammar and tried to address all the issues pointed out. Please find the response to each point below. Your original comment is followed by our response in red.

  1. The introduction of the article contains too much content; the author needs to reduce it.

Although longer than most introductory sections, we believe that contents in the introduction are essential to set the stage for the analysis, to connect Geosciences and Economics analysis and to provide a basis for the reader that may not be familiar with the problem at hand, providing the rationale for the analysis that follows. We believe that reducing the introduction would be detrimental to the manuscript.

  1. Figure 1 is suggested for modification; the conceptual model schematic is prone to confuse readers.

Figure 1 is a simplified conceptual model by design, aiming at providing a simple ‘roadmap’ for the kind of issues that our analysis tackled. It was not meant to be a comprehensive depiction of the intricacies related to the accumulation of organic-rich sediments, something that can be found in more detail in papers such as Sageman et al. (2003)*, Sageman and Lyons (2004)**.

*A tale of shales: the relative roles of production, decomposition, and dilution in the accumulation of organic-rich strata, Middle–Upper Devonian, Appalachian basin. Chemical Geology 195: 229– 273

** Geochemistry of Fine-grained Sediments and Sedimentary Rocks. In: Treatise on Geochemistry, Volume 7. Editor: Fred T. Mackenzie. Executive Editors: Heinrich D. Holland and Karl K. Turekian. Elsevier, p.115-158

  1. The content of the experimental methods and samples requires logical organization. The author must add more subheadings to distinguish sample sources and experimental methods. For example, experimental methods include cluster analysis, Ordinary Least Squares, etc.

We have organized the different methods in sub-sections (2.1 Cluster analysis, 2.2. Discrimination and classification analysis, 3.3. Linear regressions analysis, 3.4 Sensitivity test) as suggested.

  1. In the experimental methods, the description of the least squares method is overly detailed and lengthy; a more concise presentation is recommended.

The detailed description of this method aims at providing the readers with the fundamental concepts required to follow our analysis and to evaluate our interpretations, even if they are not particularly well versed in Statistics. We appreciate that it may be too detailed for those with strong statistical skills, but we are trying to reach a broader audience that could perhaps apply the same approach, not only validating but also improving and expanding our analysis. Our manuscript is meant to be an exercise, hopefully serving as a building block for future analyses.

  1. Figure 3 to 5 needs to be redrawn by the author; the relevant coordinates are not clearly presented.

We modified the figures, increasing the fonts, as suggested.

  1. The discussion and experimental results of the article need to be separated. The author needs to reduce the research content and rewrite subheadings. The word count in Manuscript 3.2 is excessively long.

We chose to present the results of each analysis and discuss them immediately after (rather than in the traditional fashion of complete separation between results and discussion) for two reasons: one, to facilitate the interpretation of the results for a readership of geoscientists unfamiliar with the statistical methods, attempting to ‘translate’ what these statistical results meant; and two, to avoid repetition of what results consubstantiate which interpretation. In other words, we thought it would be easier for the reader and shorter to do it in this fashion.

We separated section 3.2 into different sections (3.2 Cluster analysis and 3.3. linear regression analysis).